# Reservoir Adaptability Evaluation and Application Technology of Carbon Quantum Dot Fluorescent Tracer

**Jinjian Chen** [1,2] , **Jianxin Liu** [1,2,*], **Jijian Dai** [3], **Bo Lin** [3], **Chunyu Gao** [1,2] **and Ci Wang** [1,2]

[1] School of Petroleum Engineering, Yangtze University, Wuhan 430100, China
[2] Key Laboratory of Drilling and Production Engineering for Oil and Gas, Wuhan 430100, China
[3] Oil Production Plant No. 1, Sinopec Henan Oilfield Branch, Nanyang 474780, China
**\*** Correspondence: liujianxin@yangtzeu.edu.cn

**Abstract:** This study investigates the application of carbon quantum dots as tracers in inter-well connectivity monitoring. A new laboratory-made water-soluble carbon quantum dot fluorescent tracer (CQD-W) was studied using 3D fluorescence characterization, structural characterization, reservoir suitability evaluation, and core flow experiments. The experimental results showed that CQD-W has a size of about 2 nm, a minimum detection limit of $10^{-2}$ mg·L$^{-1}$. It has good stability when the salinity is 200,000 mg·L$^{-1}$, the concentration of Ca$^{2+}$ is 1000 mg·L$^{-1}$, the pH value is 1–9, and the temperature is 80 °C. Because CQD-W contains many functional groups, such as carboxyl and hydroxyl, it shows good water solubility and has a negative surface charge. In the process of formation flow, CQD-W has a small adsorption amount, high tracer resolution, and excellent injectivity and mobility, meaning it is less likely to cause reservoir damage. Through the study of this method, the application field of carbon quantum dots is broadened, and it is proved that the CQD-W fluorescent tracer has a high potential for application in the oil industry, laying the foundation for the popularization of this technology.

**Keywords:** carbon quantum dots; fluorescent tracers; nanoparticles; reservoir adaptation

## 1. Introduction

In oil and gas development, it is important to obtain inter-well parameters. Tracer technology is considered an effective way to obtain subsurface information [1,2]. The tracers currently used are primarily classified as radioactive tracers and non-radioactive tracers [3,4]. Radioactive tracers are primarily represented by tritium and tritium compounds that are used in small quantities, are easy to detect, and have a minimum detection limit of $10^{-5}$ mg·L$^{-1}$. However, they are limited due to their radioactivity, the complexity of operation, hazards to staff, and environmental safety [5]. The non-radioactive tracers are primarily thiocyanates (SCN$^{-}$), halogenated ions (Cl$^{-}$, Br$^{-}$), fluorescent dyes, halogenated hydrocarbons, and alcohols. They have been commonly used as inter-well tracers due to their wide variety and low toxicity. However, these tracers also have disadvantages, such as poor adaptability and selectivity, easy interaction with minerals, and adsorption [6–9].

Carbon quantum dots are a new class of carbon nanomaterial with fluorescent properties [10,11]. Due to their advantages in water solubility, chemical inertness, low toxicity, easy functionalization, and resistance to photobleaching, they are primarily used in biomedical imaging [12–17], photocatalysis [16–19], and water treatment [20,21]. These nanomaterials consist of a carbon core (internal carbon atoms) and surface functional groups. The carbon core is a skeleton composed of sp$^2$ heterogeneous graphitic microcrystalline carbon and sp$^3$ heterogeneous amorphous carbon. Its surface usually shows abundant oxygen-containing functional groups, such as hydroxyl and carboxyl groups [22–29]. Therefore, the carbon quantum dots exhibit good stability and water solubility. In addition, a rigid planar structure is formed in the center of the carbon nucleus. The energy trap formed on the surface can trap additional electrons to ensure that it fluoresces strongly when excited [25,30].

Related studies in the biomedical field have shown that carbon quantum dots exhibit strong mobility and have more advantages in terms of luminescence intensity and stability than conventional organic dyes [31,32]. In addition, these nanomaterials inherit the biocompatibility of carbon materials and can meet the needs of real-time monitoring of complex biological environments [33,34].

In recent years, related research scholars started to study the application of carbon quantum dots as tracers in oil fields. Murugesan et al. [2] synthesized a nitrogen-doped carbon quantum dot using an electrochemical method and preliminarily verified the potential of carbon quantum dots as a tracer through a core flow experiment. Ma et al. [1] synthesized five carbon quantum dots using a hydrothermal method. He conducted temperature and acid-base resistance tests as well as core flow experiments. He confirmed that the adsorption of carbon quantum dots on the rock surface determines whether they can be effective as tracers. Hu et al. [35] synthesized a carbon quantum dot and studied its flow performance in sandstone cores. The results showed that the carbon quantum dots could detect the oil saturation of unknown sandstone cores. Shi et al. [25] synthesized a high fluorescence yield quantum dot using a hydrothermal method and confirmed that carbon quantum dots have good stability, mobility, and low adsorption with different minerals through reservoir adaptability evaluation experiments and core flow experiments.

Previous studies on applying carbon quantum dots as tracers in the petroleum industry have focused on the reservoir suitability study and oil saturation determination of carbon quantum dots. In this paper, the team has designed a Si-doped CQD-W fluorescent nanotracer, conducted preliminary research on its three-dimensional fluorescence characteristics, structural characteristics, and reservoir adaptability, and verified the injectivity and mobility of CQD-W through single-tube core flow experiments. Finally, a double-tube core flow experiment study was conducted to confirm whether carbon quantum dots have the ability to distinguish dominant channels. This latter experiment is expected to be applied to oil field development.

## 2. Materials and Methods

### 2.1. Experimental Materials

Main experimental reagents used are showned in Table 1.

**Table 1.** Main experimental reagents.

| Reagent Name | Purity | Manufacturers |
|---|---|---|
| Sodium chloride (NaCl) | Analysis of pure | National Pharmaceutical Reagent |
| Calcium chloride (CaCl$_2$) | Analysis of pure | Sinopharm Reagent |
| Disodium hydrogen phosphate (Na$_2$HPO$_4$) | Analysis of pure | National Pharmaceutical Reagent |
| 37% Hydrochloric acid (HCl) | Analysis of pure | National Pharmaceutical Reagent |
| Citric acid anhydrous (C$_6$H$_8$O$_7$) | Analysis of pure | National Pharmaceutical Reagent |
| Potassium bromide (KBr) | Analysis of pure | National Pharmaceutical Reagent |

CQD-W is homemade in the laboratory; quartz sand is purchased from Jiangxi Shanggao Huasil Mining Co.; Artificial Sandstone Cores ($\Phi 2.5 \times 10$) A (611 mD) and B (795 mD).

### 2.2. Experimental Methods

#### 2.2.1. Method of Infrared Spectroscopy Test

6 mg of CQD-W and 600 mg of KBr were taken and ground, and 10 Mpa pressure was applied, pressing the materials into thin slices. The treated samples were tested by infrared spectroscopy using a Fourier transform infrared spectrometer (Vector-33, Bruker Instruments, Germany) [1,35].

#### 2.2.2. Method of ζ-Potential Test and Particle Size Characterization

A ζ-potential cuvette was loaded with 1 mg·L$^{-1}$ CQD-W solution 0.5 mL and the cuvette was placed into a nanoparticle size ζ-potential analyzer (Litesizer 500, Anton Paar,

Austria) for $\zeta$-potential testing. The CQD-W particle size testing was performed similarly, with a quartz cuvette chosen for the test [36].

### 2.2.3. Method of Fluorescence Characteristics Test

The concentration of 1 mg·L$^{-1}$ CQD-W solution was configured. 1 mL of the solution was taken in a quartz fluorescent cuvette and the quartz fluorescent cuvette was put into a fluorescence spectrophotometer (F97pro, Shanghai Prism, China) for three-dimensional fluorescence scanning with a slit width of 10 and a gain of 9 (the fluorescence intensity tests were all tested under this parameter).

The standard concentration solutions of 0~20 mg·L$^{-1}$ were configured. The quartz fluorescent cuvettes with a standard concentration of CQD-W solutions were put into a fluorescence spectrophotometer (F97pro, Shanghai Prism, China) at the optimal excitation wavelength, emission wavelength, and slit width to measure the fluorescence intensity. Finally, the relationship between CQD-W and fluorescence intensity was obtained.

### 2.2.4. Method of Reservoir Suitability Evaluation

The CQD-W mixed solutions with concentrations of 0~200,000 mg·L$^{-1}$ NaCl of 0.1 mg·L$^{-1}$ CQD-W, 0~1000 mg·L$^{-1}$ CaCl$_2$ of 0.1 mg·L$^{-1}$ CQD-W, and 0.1 mg·L$^{-1}$ CQD-W at pH = 1~9 were prepared and tested for CQD-W fluorescence intensity by diluting 10 times with pure water.

CQD-W mixed solutions with concentrations of 0.1 mg·L$^{-1}$ CQD-W and 5000 mg·L$^{-1}$ NaCl were prepared, placed in small sample bottles, sealed and heated in an electric thermostatic water bath (DK-8AX, Shanghai Heheng Instruments, China). Every 24 h, a portion of the mixed solution was removed and diluted 10 times using pure water for the CQD-W fluorescence intensity test (repeated for one week).

$$\delta = \frac{x_1 - x_2}{x_1} \tag{1}$$

where: $\delta$—relative error.

$x_1$—control CQD-W concentration.

$x_2$—CQD-W concentration in the experimental group.

The CQD-W mixed solution with concentrations of 0.1 mg·L$^{-1}$, 5000 mg·L$^{-1}$ NaCl was prepared, and the 0.1 mg·L$^{-1}$ CQD-W solution and liquid paraffin were mixed according to the volume ratio (20:20) with magnetic stirring for 5 h. Then, the solution was transferred to a partition funnel and left for 2 h. After being completely layered, the aqueous/oil phase fraction was removed for fluorescence intensity test analysis [37].

### 2.2.5. Static Adsorption Experiments

CQD-W solution of 0.1 mg·L$^{-1}$ was added to a 250 mL reagent bottle with quartz sand (60–120 mesh) at a solid-liquid ratio of 1:4 (mass ratio). Then the bottle was placed in a 50 °C digital water bath shaker (SHA-CA, Changzhou Jintan Kexing Instruments) for 2 h. Every 2 h, all the supernatant samples were removed and separated by centrifugation at 4000 r/min for 10 min in a benchtop high-speed centrifuge (TG16, Shanghai Lu Xiangyi Centrifuge Instruments Co. Ltd.). Part of the supernatant was removed to determine its fluorescence intensity, then new quartz sand was added according to the solid-liquid ratio of 1:4 for adsorption. This process was repeated 8 times [38].

$$\Gamma = \frac{(C_0 - C_1)V}{m} \tag{2}$$

where: $\Gamma$—apparent adsorption of CQD-W per unit mass of quartz sand, ug/g.

$C_0$—concentration of CQD-W before adsorption, mg/L.

$C_1$—concentration of CQD-W after adsorption, mg/L.

$V$—the volume of CQD-W added, mL.

$m$—single quartz sand mass, g.

## 2.2.6. Core Flow Experiments

The multi-functional core repelling device (Jiangsu Lianyou Research Instruments) was installed. The core A (porosity 25.1%) was saturated with deionized water before the experiment and injected with a mixture of 5000 mg·L$^{-1}$ NaCl and 1 mg·L$^{-1}$ CQD-W solution at a constant flow rate of 1 mL/min at standard atmospheric pressure and a temperature of 25 °C. The brine concentration of the extracted fluid and the concentration of the CQD-W solution were measured periodically for a certain PV. The concentration output curves of NaCl and CQD-W were summarized. The schematic diagram of the single-tube repulsion device is shown in Figure 1.

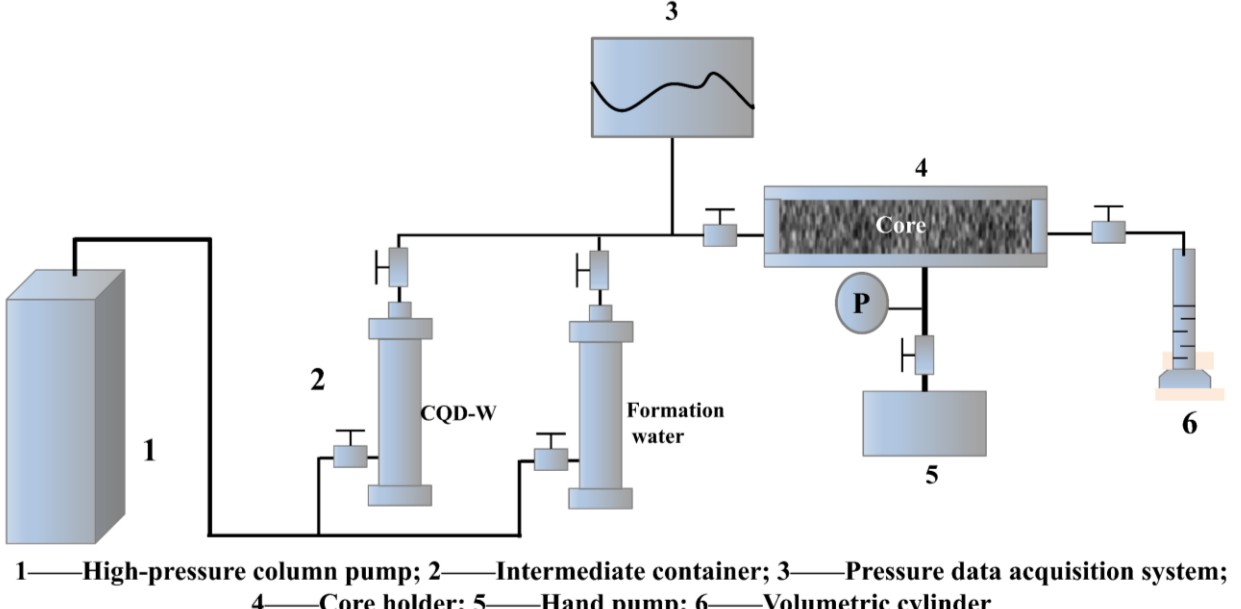

1——High-pressure column pump; 2——Intermediate container; 3——Pressure data acquisition system; 4——Core holder; 5——Hand pump; 6——Volumetric cylinder

**Figure 1.** Schematic diagram of core flow device.

The experimental method for the double-tube core flow is similar, except that an additional group of core B (porosity 30.4%) is connected in parallel for the experiment (cores A and B are saturated with 5000 mg·L$^{-1}$ NaCl before the experiment), and 0.1 PV 5000 mg·L$^{-1}$ NaCl mixed with 5 mg·L$^{-1}$ CQD-W is injected at a constant flow rate of 2 mL/min. The concentration is detected as above.

## 3. Results and Discussion

### 3.1. Characterization Analysis of CQD-W

3.1.1. Infrared Spectral Analysis of CQD-W

As shown in Figure 2, the Fourier transform infrared (FTIR) spectra indicated CQD-W exhibits abundant hydrophilic groups on its surface, including O-H (3600–3100 cm$^{-1}$), C=C (1525.63 cm$^{-1}$), C=O (1639.03 cm$^{-1}$) and Si-O (1208.87 cm$^{-1}$), thus promoting a good solubility in water. In addition, the O-H vibrational band, about 3437.14 cm$^{-1}$, is broad peaked, indicating that the CQD-W surface has multiple structures of hydroxyl groups, thus leading to high polarity and hydrophilicity [35,39,40]. These functional groups make the carbon quantum dots scalable and their surface tunable for a variety of applications [7,41,42].

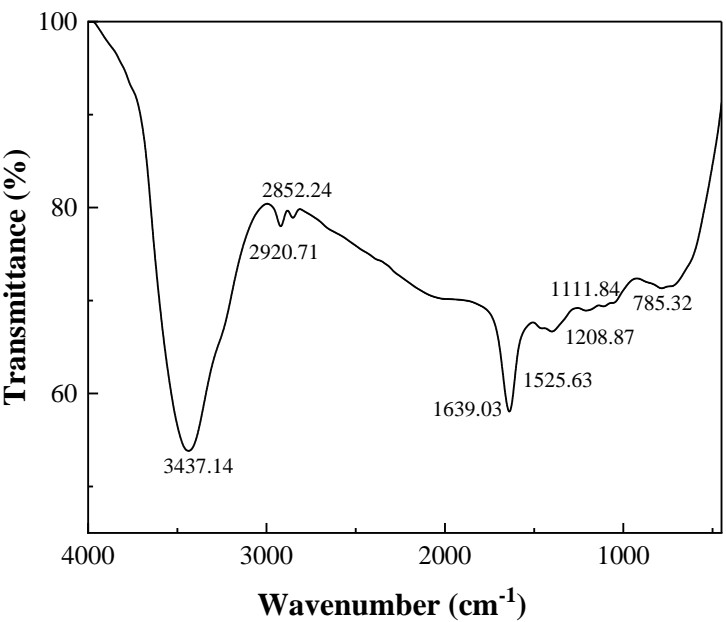

**Figure 2.** Fourier Transform Infrared (FTIR) Spectroscopy of CQD-W.

### 3.1.2. ζ-Potential Analysis of CQD-W

As shown in Figure 3, the average ζ-potential of CQD-W is 0.3 mV, primarily due to the presence of Si-O and carboxyl groups' in the CQD-W structure. In general, the carboxyl group exists in the dissociated state (-COO-). The covalent bond between Si-O is very polar, and it will attract many hydrogen ions in water to form Si-OH, that will be ionized under certain conditions [43–47].

$$Si - OH \rightleftharpoons Si - O^- + H^+ \tag{3}$$

$$Si - OH + OH^- \rightleftharpoons Si - O^- + H_2O \tag{4}$$

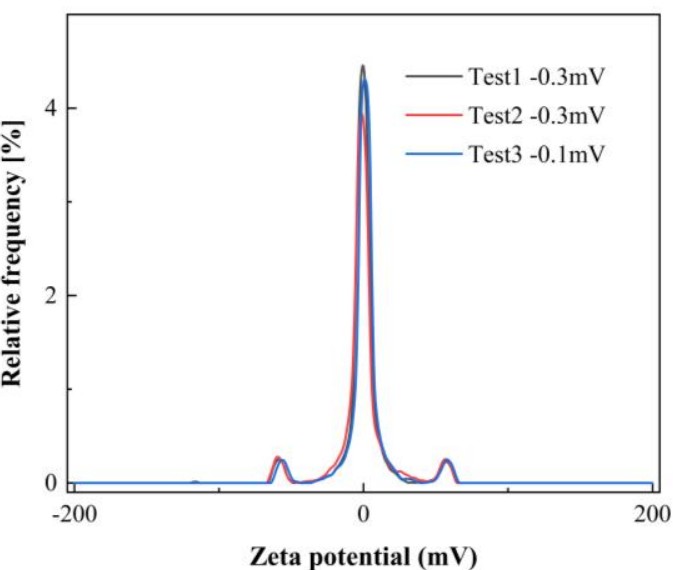

**Figure 3.** ζ-potential distribution of CQD-W.

### 3.1.3. Particle Size Test Analysis of CQD-W

As shown in Figure 4, CQD-W has a very narrow strong peak at 0~2 nm, indicating that CQD-W has good particle size and particle size homogeneity (particle size is about

1 nm). The size of CQD-W microspheres needs to be adapted to the fluid flow in the formation. According to the equation of the hydrodynamic radius of the pore throat [48]:

$$R = \frac{d}{4} = \frac{r}{2} = \frac{1}{2}\left(\frac{8K}{\phi}\right)^{0.5} \tag{5}$$

where: *R*—the hydrodynamic diameter of the orifice throat.

    *K*—permeability.

    $\phi$—Take the value of 10%.

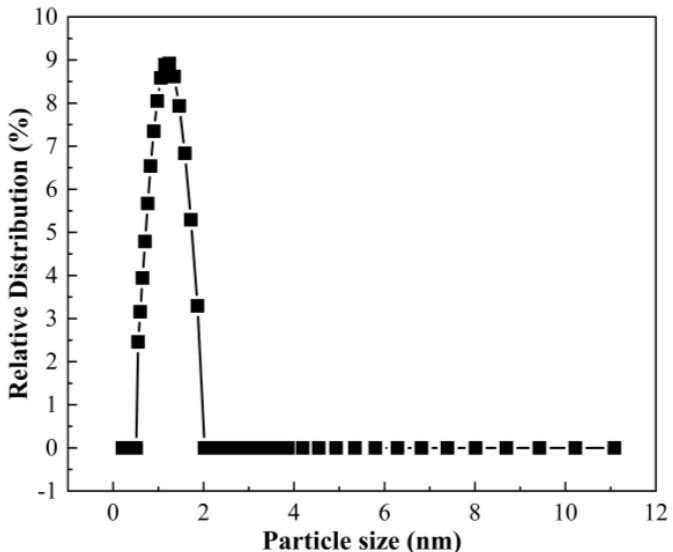

**Figure 4.** Particle size distribution (intensity) diagram of CQD-W.

To ensure that the tracer particles can pass through the formation pore throat smoothly, the radius of the tracer microspheres should be no larger than the hydrodynamic radius of the pore throat. For size 10 nm particles, this generally applies to reservoirs with permeability above 0.005 mD.

*3.2. Analysis of CQD-W Fluorescence Characteristics*

3.2.1. Three-Dimensional Fluorescence Characterization of CQD-W

As shown in Figure 5, CQD-W appears clear and transparent under natural light conditions and exhibits a strong fluorescence emission band (green fluorescence) under UV light irradiation.

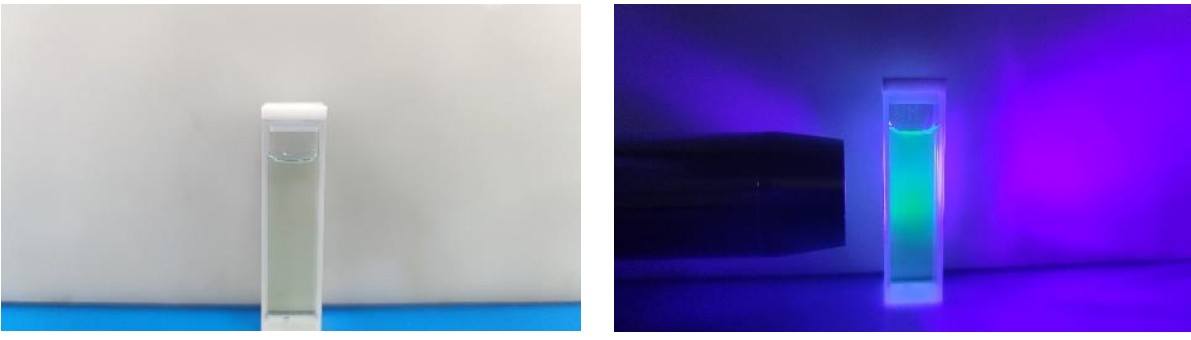

**Figure 5.** Color development of CQD-W under natural light (**left**) and UV light (**right**) irradiation.

As shown in Figure 6, the three-dimensional fluorescence spectra showed that the fluorescence waveform of CQD-W had a single fine shape and obvious fluorescence charac-

teristics with good discrimination. The best excitation wavelength was 491.1 nm and the best emission wavelength was 518.9 nm.

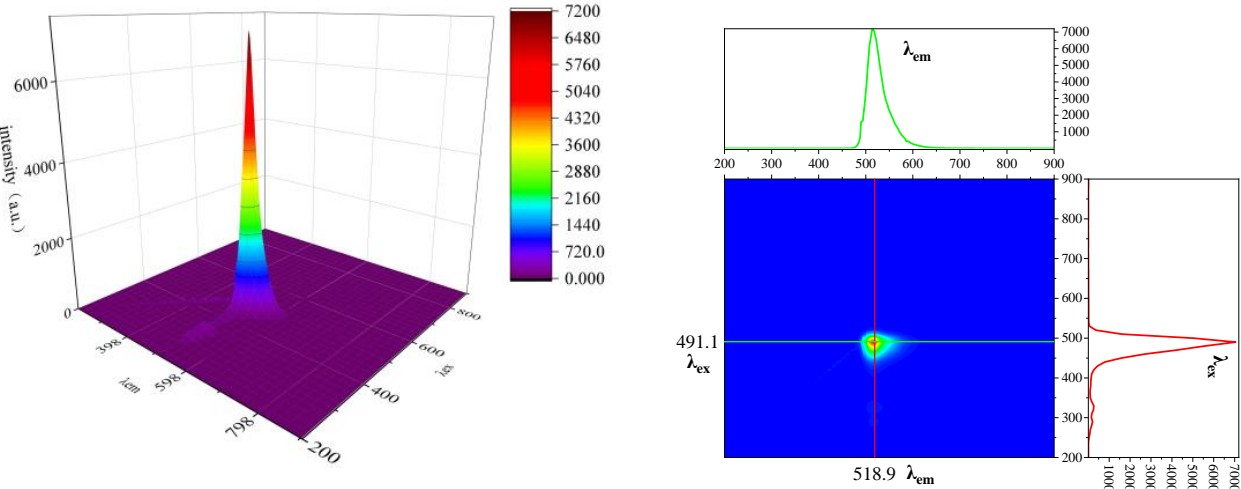

**Figure 6.** 3D scanning situation of CQD-W (**left**) top view profile of CQD-W (**right**).

### 3.2.2. Standard Curve of CQD-W

As shown in Figure 7, the concentration of CQD-W is no longer linear above 3 mg·L$^{-1}$. This is primarily because when the CQD-W concentration is too high due to factors such as internal filtration and energy transfer, causing the fluorescence intensity and concentration are not stable, and fluorescence quenching phenomena such as linearity and spectral distortion [49–52]. In the concentration range of 0 mg·L$^{-1}$ to ~0.5 mg·L$^{-1}$, the tracer concentration has an excellent linear correlation with the fluorescence intensity (bandwidth 10, gain 9), which satisfies the Lang–Bier law. It can be used as a standard curve for concentration measurement [53,54], and its linearity equation is as follows.

$$I = -55.50 + 13697.96 \times C \ \left(R^2 = 0.998\right) \tag{6}$$

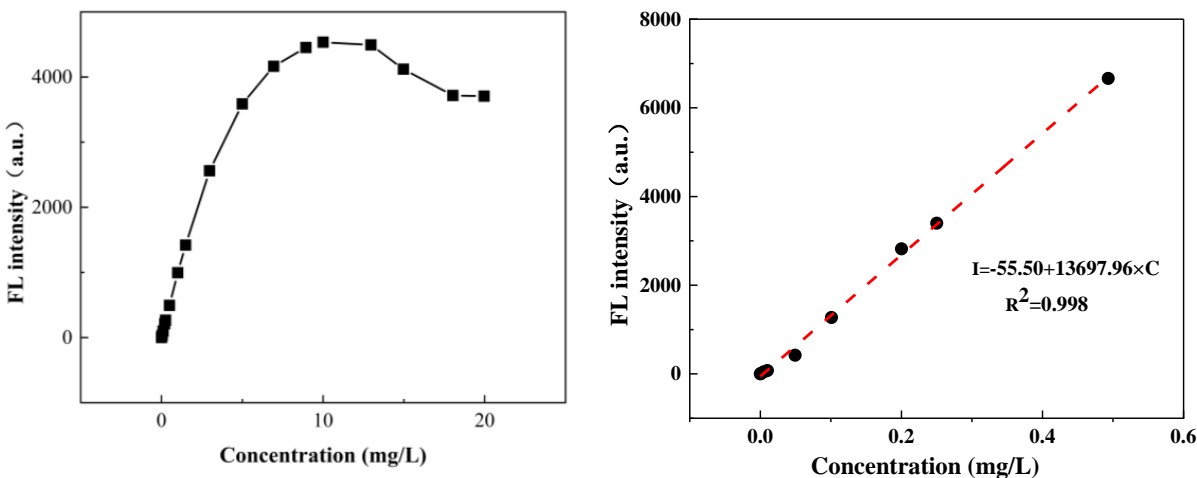

**Figure 7.** The concentration of CQD-W versus fluorescence intensity (bandwidth 10, gain 7).

### 3.3. Reservoir Suitability Analysis of CQD-W

3.3.1. Stability Analysis of CQD-W

Common ions in stratigraphic water may bind to the tracer and thus alter the fluorescence intensity of the tracer [55]. This paper examines the measurements of 0~200,000 mg·L$^{-1}$ NaCl solution and 0~1000 mg·L$^{-1}$ CaCl$_2$ solution in a calibrated 0.1 mg·L$^{-1}$ tracer solution.

As shown in Table 2, the mineralization and Ca$^{2+}$ had minimal interference with the concentration of the tracer and their concentration deviations were within 5%, indicating that the CQD-W tracer has good stability [25,35,56].

**Table 2.** Effect of stratigraphic ions on the stability of CQD-W.

| The Concentration of NaCl (%) | The Concentration of CQD-W (mg/L) | Relative Error | The Concentration of CaCl$_2$ (mg/L) | The Concentration of CQD-W (mg/L) | Relative Error |
|---|---|---|---|---|---|
| Blank group | 0.10083 | 0.00% | Blank group | 0.10307 | 0.00% |
| 0.5 | 0.10087 | 0.04% | 50 | 0.10482 | 1.70% |
| 5 | 0.10206 | 1.21% | 100 | 0.10488 | 1.76% |
| 10 | 0.10243 | 1.58% | 500 | 0.10581 | 2.66% |
| 20 | 0.10190 | 1.06% | 1000 | 0.10693 | 3.75% |

The concentration variation of CQD-W solution of 0.1 mg·L$^{-1}$ in the temperature range of 30~80 °C is shown in Figure 8. The results showed that temperature and aging time influenced the measurement results, and its measurement error was basically within 10%. The maximum deviation did not exceed 8%, indicating that CQD-W has good reservoir adaptability [25,35].

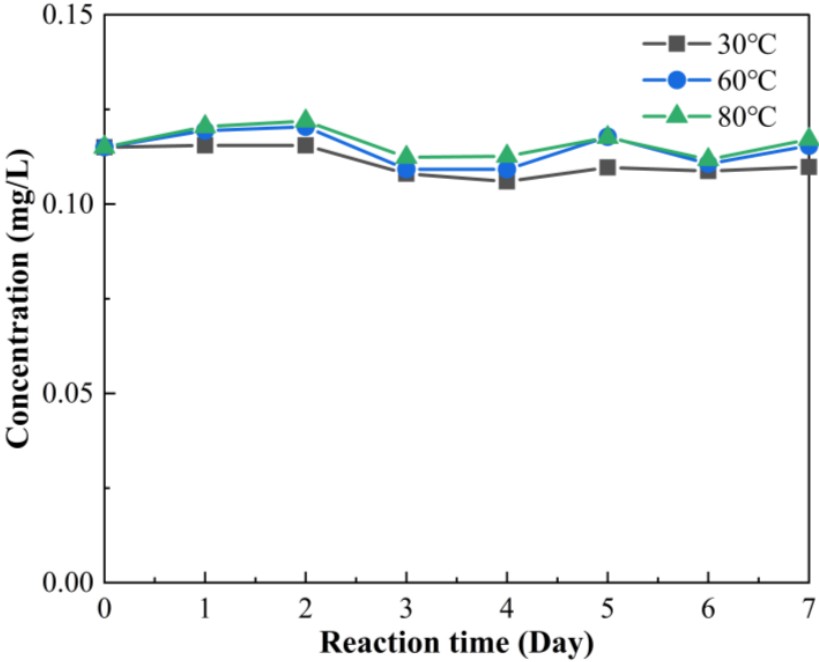

**Figure 8.** Changes in concentration of CQD-W after aging at different temperatures over a period of time.

As shown in Table 3, the pH value of the solution had little effect on the measurement results, and its concentration deviation was within 6%, indicating that the tracer has good pH stability [56].

**Table 3.** Effect of pH on concentration test of CQD-W.

| pH | The Concentration of CQD-W (mg/L) | Relative Error |
|---|---|---|
| Blank group | 0.10785 | 0.00% |
| 1 | 0.10189 | 5.53% |
| 3 | 0.10234 | 5.11% |
| 5 | 0.10714 | 0.66% |
| 7 | 0.10416 | 3.43% |
| 9 | 0.10564 | 2.05% |

### 3.3.2. Solubility of CQD-W in Oil and Water

Since formation fluids generally exist as multiphase flows with oil and water coexistence, the dissolution properties and partitioning ratio of tracer in oil and water phases significantly impact the separation methods and test results of samples [57,58]. As shown in Figure 9, the water phase part of the solution showed obvious fluorescence under UV irradiation after CQD-W and liquid wax were entirely mixed and left to stand. In contrast, the oil phase part did not show fluorescence characteristics.

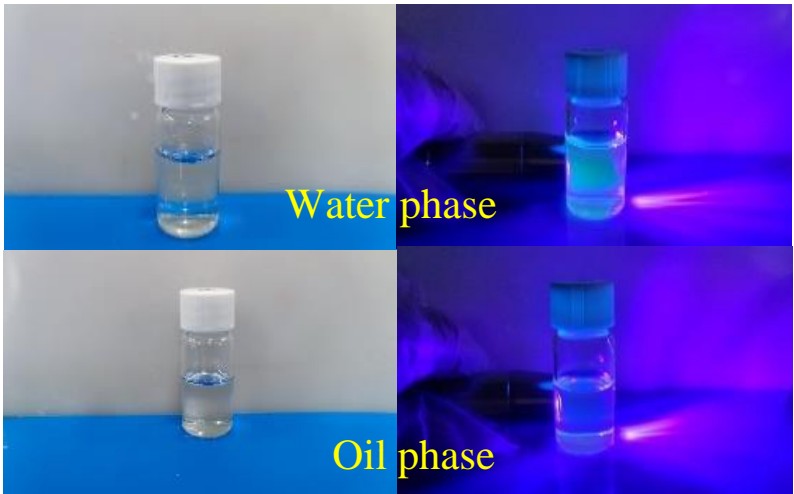

**Figure 9.** Solution after static stratification.

As shown in Figure 10, the fluorescence value of CQD-W changed very little after oil-water mixing, indicating that this CQD-W has good hydrophilic properties. It is because CQD-W contains many carboxyl and hydroxyl groups, that can easily form hydrogen bonds with water molecules and therefore has better water solubility [59,60]. In the actual testing process, only a simple oil-water separation of the extracted fluid is required for testing and analysis.

### 3.3.3. Static Adsorption Experiments of CQD-W

Tracer adsorption occurs when the tracer flows over the rock surface. This leads to tracer loss and may impact the tracer concentration detection during the actual detection process. As shown in Figure 11, the tracer concentration of CQD-W was 0.92 mg·$L^{-1}$ after eight repeated adsorptions, with a fluctuation of 8.7% relative to the initial concentration and average adsorption of 0.0435 ug/g. This is because CQD-W is negatively charged with the formation rock and its adsorption is low under the effect of electrostatic repulsion [25].

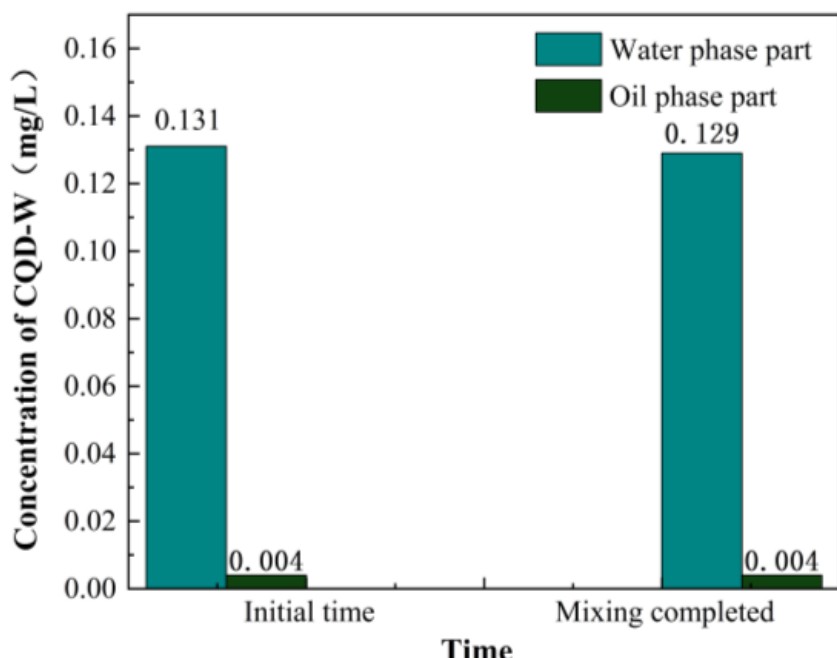

**Figure 10.** Change in concentration of CQD-W after static stratification of the solution.

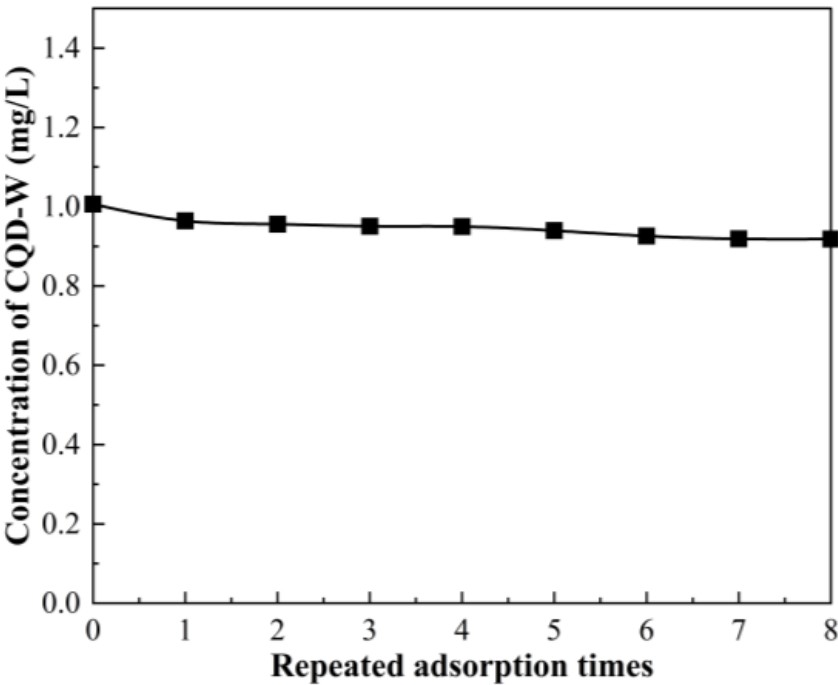

**Figure 11.** The concentration of CQD-W with the number of repetitions of adsorption.

### *3.4. Core Flow Experiment*

3.4.1. Injectivity and Brine Flow Variability Analysis of CQD-W

As shown in Figure 12, in CQD-W and brine synchronous flow experiments, CQD-W and brine flow were consistent with each other and had good flow performance inside the core. No adsorption hysteresis was observed in the CQD-W concentration curve when compared with the brine flow curve. In the process of core flow, the pressure inside the core remained unchanged after injecting CQD-W and pure water. It indicates that the internal pore structure of the core was not changed after the flow of CQD-W, which is not easy to cause reservoir damage and other problems [25].

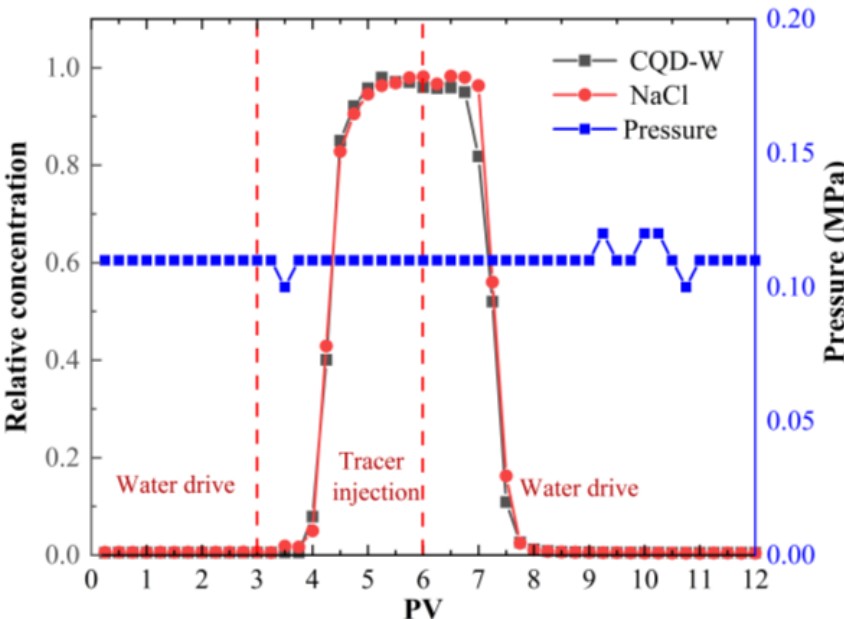

**Figure 12.** Relative concentration and internal pressure versus injection volume.

As shown in Figure 13, the cumulative recovery of NaCl reached 99.21%, and the cumulative recovery of CQD-W reached 96.83%. The two confirm each other, indicating that the adsorption between CQD-W and the core is minimal, primarily because of the repulsive effect between CQD-W and the negatively charged core [7,25].

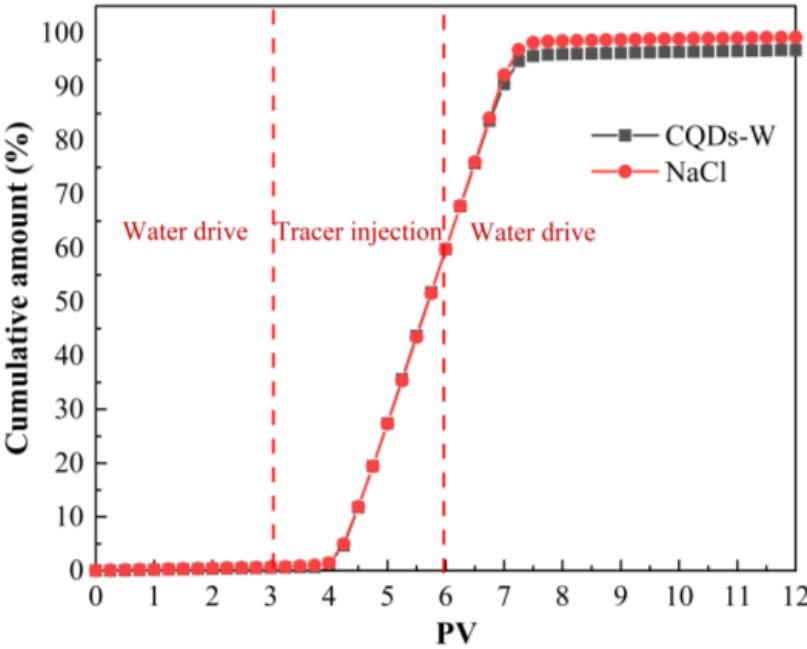

**Figure 13.** CQD-W cumulative volume versus injection volume.

### 3.4.2. Description of Cores by CQD-W

As shown in Figure 14, after injecting a 0.1 PV small segment plug CQD-W in two segments of the core with little difference in permeability, the slightly larger permeability core saw the CQD-W first. This is because the flow rate of the segment plug in the flow pipe increases after the permeability increases and the time to see the CQD-W becomes shorter. With the same plug length, the time difference between the leading and trailing

edges of the plug to reach the production well becomes shorter. Therefore, the overall time to see the CQD-W becomes shorter. This is reflected in the shape of the curve, i.e., the bandwidth becomes narrower [61]. As a side effect, the CQD-W tracer can distinguish superior channels with higher permeability and has excellent tracer resolution.

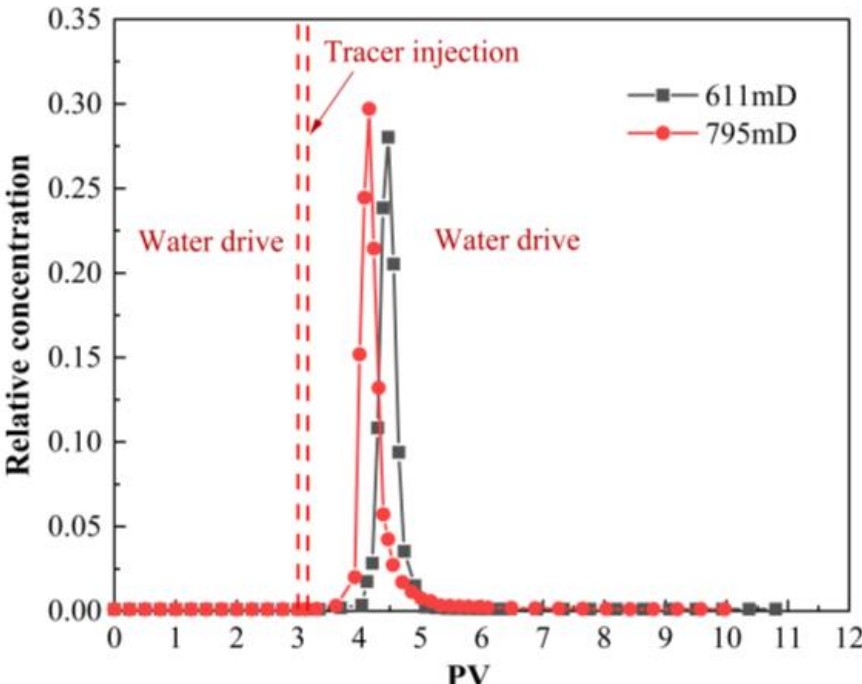

**Figure 14.** Relative concentration versus injection volume.

As shown in Figure 15, the cumulative recovery of low and high permeability cores reached 97.41% and 96.57%, respectively, indicating that the adsorption of CQD-W is low and meets the performance requirements for the tracer used.

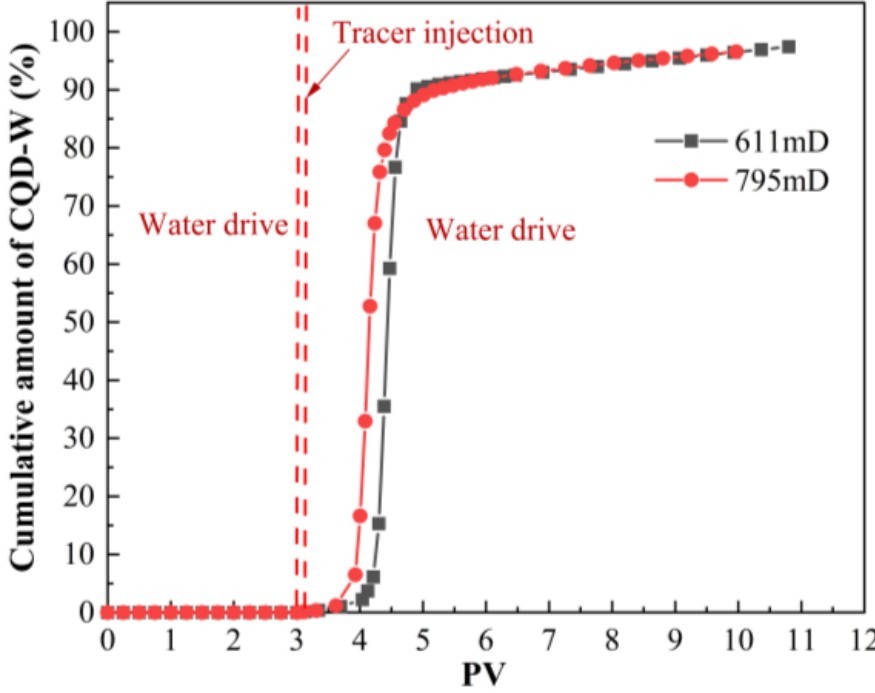

**Figure 15.** The cumulative volume of CQD-W versus injection volume.

As shown in Figure 16, the two sections of cores with little difference in permeability were subjected to core flow experiments. The shunt ratios of both in the pre-water drive, CQD-W injection, and subsequent water drive stages were maintained, indicating that the internal pore structure of the cores was less affected by the flow of CQD-W [62,63].

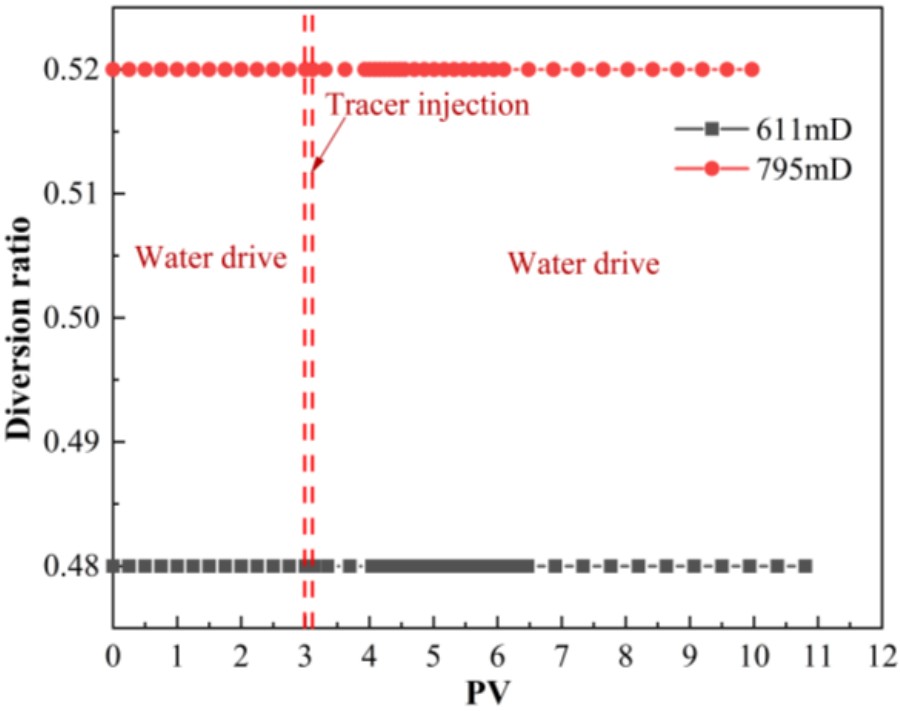

**Figure 16.** Relationship between shunt ratio and injection volume.

## 4. Conclusions

The surface of CQD-W is rich in polycarboxyl and siloxy groups with good water solubility, and CQD-W shows negative electrical properties in pure water; this makes CQD-W adsorption on the rock surface very low. The particle size characterization results show that the average particle size of CQD-Ws is 2 nm; this can meet the reservoir mobility and injectivity with permeability above 0.005 mD.

(1)  CQD-W three-dimensional fluorescence spectrogram shows that the three-dimensional fluorescence peak is a single peak that has better differentiation in practical application detection. The minimum detection limit of CQD-W reaches $10^{-8}$. CQD-W has the advantage of small dosage, convenient detection, and the luminescence situation of CQD-W is very little affected by temperature, mineralization, and pH.

(2)  In the core flow experiment, after injecting a 0.1 PV small section plug CQD-W tracer in two sections of cores with little difference in permeability, the permeability of slightly larger cores saw the CQD-W first. This shows that CQD-W can distinguish the superior channel with higher permeability and has excellent tracer resolution. At the same time, CQD-W has an excellent flow ability and will not change the internal pore structure of the core during the flow, making it more difficult to produce reservoir damage. The cumulative recovery rate in the low and high permeability cores reached 97.41% and 96.57%, respectively, primarily because CQD-W has negative electricity and the negative core has a repulsive effect between the two.

(3)  CQD-W testing and characterization, reservoir suitability analysis, and core flow experiments can verify that CQD-W meets the performance requirements of conventional tracers and, to a certain extent, proves the possibility of CQD-W's future application in oil field sites. It can also provide in-depth guidance on the preparation of target CQD-W materials and broaden the application fields of carbon quantum dots.

**Author Contributions:** Conceptualization, J.L.; data curation, J.C.; formal analysis, J.C. and C.W.; methodology, J.C. and C.G.; project administration, J.C. and J.D.; resources, J.L. and B.L.; supervision, J.L. and B.L.; writing—original draft, J.C.; writing—review and editing, J.L. All authors have read and agreed to the published version of the manuscript.

**Funding:** This research was supported by the Open Foundation of Hubei Key Laboratory of Oil and Gas Drilling and Production Engineering, Yangtze University (No. YQZC202105) and the Yangtze University 2021 Innovation and Entrepreneurship Training Program for College Students "Research on the Mechanism of Deep Drive Regulating by Resistant Polymer Microspheres" (Project No. Yz20211253).

**Institutional Review Board Statement:** Not applicable.

**Informed Consent Statement:** Not applicable.

**Data Availability Statement:** Not applicable.

**Acknowledgments:** We would like to thank the Key Laboratory of Drilling and Production Engineering for Oil and Gas of Yangtze University for their support. At the same time, I would like to thank my teacher, Jian-xin Liu, for his help.

**Conflicts of Interest:** The authors declare no conflict of interest.

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
