# Peer review of "Reservoir Adaptability Evaluation and Application Technology of Carbon Quantum Dot Fluorescent Tracer"

_2673-4117, doi:10.3390/eng4010042_

Round 1
Reviewer 1 Report
Comments as per file attached.

Reviewer 2 Report
This manuscript describes a carbon quantum dot compound and demonstrates its suitability as a reservoir tracer by batch tests and core flooding experiments. There are a number of missing information and inconsistencies in the manuscript which should be addressed before further consideration:
General comments:
Missing information in methods section:
- Section 2: Synthesis procedure of CQD-W was not mentioned other than it was “homemade”. Please provide the synthesis protocol for the nanoparticle tracer, or at least a reference where readers can find it.
- What are the dimensions of the cores/sand packs? How much is 1 PV for each porous medium? How were the permeability values and pore volumes measured? I recommend adding a table summarizing the above information.
Is scanning electron microscope (SEM) imaging of CQD-W available? SEM is a nice way to help readers picture what the particles look like.
Lines 136-138: Why did you add fresh sand after each removal of supernatant? How can this be representative under real reservoir conditions?
Section 3.1.2: It is unclear how the state of silanol groups is relevant to the zeta potential of carbon dots. Does your carbon dots contain silica?
Lines 285-287: This statement is contradictory to the statement in line 153 that the core flooding experiment was conducted at a constant 2 ml/min.
Lines 290-291: “… the CQD-W tracer can distinguish superior channels with higher permeability and has excellent tracer resolution”: This can be done by conventional tracers too, such as Br- and Cl-, right? How is CQD-W better?
Figure 16: What is “diversion ratio” on the y axis? What is “shunt ratio” in the caption and in line 296? Please define.
Can the synthesis of CQD-W be upscaled? Please add discussion on the advantages and challenges for applying CQD-W in field-scale tracer tests.
Specific comments:
- Lines 17, 34, etc.: Please provide units for the detection limit.
- Lines 60, 61, etc.: Better to put it this way: “Ma et al. [provide reference] synthesized five carbon quantum dots…”, “He et al. [1] conducted…”. This applies to several other instances in the manuscript as well.
- Line 132: Please specify whether “1:4” is volumetric or mass ratio or something else.
- Line 141: How is “adsorption equilibrium” defined?
- Line 143: Is this “mass of quartz sand” cumulative? If so, please clarify.
- Line 172: Typo? Carboxyl groups is not Si-O.
Round 2
Reviewer 2 Report
The reviewers have satisfactorily addressed my concerns in the initial review. The manuscript is now publishable.